

# Snapshot recordings provide a first description of the acoustic signatures of deeper habitats adjacent to coral reefs of Moorea

Frédéric Bertucci[1,2], Eric Parmentier[2], Cécile Berthe[1], Marc Besson[1,3], Anthony D. Hawkins[4], Thierry Aubin[5] and David Lecchini[1,6]

[1] USR 3278 CRIOBE, PSL Research University, EPHE-UPVD-CNRS, Moorea, French Polynesia
[2] Laboratoire de Morphologie Fonctionnelle et Evolutive, AFFISH-RC, Université de Liège, Liège, Belgium
[3] Observatoire Océanologique de Banyuls-sur-Mer, UMR 7232, Université Pierre et Marie Curie (Paris VI), CNRS, Banyuls-sur-Mer, France
[4] Loughine Marine Research, Aberdeen, Scotland, United Kingdom
[5] Neuro-PSI, UMR 9197, Université Paris Sud (Paris XI), CNRS, Orsay, France
[6] Laboratoire d'Excellence CORAIL, EPHE, Paris, France

## ABSTRACT

Acoustic recording has been recognized as a valuable tool for non-intrusive monitoring of the marine environment, complementing traditional visual surveys. Acoustic surveys conducted on coral ecosystems have so far been restricted to barrier reefs and to shallow depths (10–30 m). Since they may provide refuge for coral reef organisms, the monitoring of outer reef slopes and describing of the soundscapes of deeper environment could provide insights into the characteristics of different biotopes of coral ecosystems. In this study, the acoustic features of four different habitats, with different topographies and substrates, located at different depths from 10 to 100 m, were recorded during day-time on the outer reef slope of the north Coast of Moorea Island (French Polynesia). Barrier reefs appeared to be the noisiest habitats whereas the average sound levels at other habitats decreased with their distance from the reef and with increasing depth. However, sound levels were higher than expected by propagation models, supporting that these habitats possess their own sound sources. While reef sounds are known to attract marine larvae, sounds from deeper habitats may then also have a non-negligible attractive potential, coming into play before the reef itself.

# INTRODUCTION

The existence of coral reefs at depths of more than 150 m in tropical regions has been known for decades (*Fricke & Schuhmacher, 1983*; *Maragos & Jokiel, 1985*; *Kahng & Maragos, 2006*). Recently, the conservation and management of these so-called mesophotic coral ecosystems (MCEs) has been considered a priority, although the reefs themselves remain largely unexplored (*Pyle et al., 2016*). More generally, deeper zones and habitats close to coral reefs may serve as refuges and be the origin of recruits that contribute to the recovery

Corresponding author
Frédéric Bertucci,
fred.bertucci@gmail.com

of reefs located in shallow waters (*Bongaerts et al., 2010*; *Van Oppen et al., 2011*). Deeper habitats were, at first, thought to be more protected from temperature increases and coral bleaching events, since the impacts of human and natural perturbations typically diminish with depth and distance from shore (*Glynn, 1996*; *Feingold, 2001*; *Glynn et al., 2001*; *Bak, Nieuwland & Meesters, 2005*). However, the current degree of global climate change may also have an impact upon deeper habitats (*Appeldoorn et al., 2016*). Very little is known about MCEs, and the deeper habitats adjacent to coral reefs. Hence, there is a wide range of possible topics to be investigated. As an example, being able to provide an acoustic description of such habitats may appear crucial for extending our current knowledge on marine soundscapes. In particular, it may provide insights into the qualities and characteristics of the deeper habitats associated with coral ecosystems (*Staaterman et al., 2014*; *Bertucci et al., 2015*; *Nedelec et al., 2015*; *Bobryk et al., 2016*).

A soundscape is defined as the collection of all sounds that are present in a landscape, which vary over space and time (*Southworth, 1969*; *Schafer, 1977*; *Krause, 1987*; *Pijanowski et al., 2011*). Within soundscapes, sound sources are divided into three main components: the biophony (corresponding to biologically produced sounds), the geophony (the geophysically produced sounds) and the anthropophony (the sounds produced by human activities). The collection of data regarding the nature and qualities of marine soundscapes is growing worldwide (*Cato & McCauley, 2002*; *Chapman & Price, 2011*; *Bertucci et al., 2016*) but despite their potential, investigations of temporal and spatial variations in the soundscapes of coral reefs have mainly been concentrated on comparing neighbouring sites consisting of different habitat types, e.g., mangrove, fringing reef and barrier reef, and they have been restricted to the first 10–20 m of the water column (*Piercy et al., 2014*; *Radford, Stanley & Jeffs, 2014*; *Bertucci et al., 2015*). For instance, *Staaterman et al. (2014)* described marine soundscapes of Florida reefs in 7 m of water while only the 0–5 m range was recorded in a French Polynesian reef by *Nedelec et al. (2015)*, and in a temperate coastal marine environment by *Rossi, Connell & Nagelkerken (2016)*. Overall, the studies highlighted that different habitat types are characterised by peculiar acoustic features that constitute their acoustic signatures. When assessing the relationship between biodiversity and soundscape features of similar reefs habitats in Virgin Islands and French Polynesia, recordings performed by *Kaplan et al. (2015)* and *Bertucci et al. (2016)* were carried out at 18 m and 10 m depth, respectively.

These studies performed in shallow waters demonstrate that acoustical differences between reef habitats are due to variations in the sonic activity of marine organisms, i.e., soniferous fishes, snapping shrimps (*Radford, Stanley & Jeffs, 2014*) and the geo-morphology of recorded sites. Acoustic cues within habitats close to coral reefs are known to influence the behaviour and orientation of many fish and invertebrate larvae at settlement (*Simpson et al., 2004*; *Vermeij et al., 2010*; *Nedelec et al., 2015*; *Parmentier et al., 2015*). Describing soundscapes from deeper habitats could further highlight the importance of acoustic cues in the distribution of marine organisms, and the attractiveness of deeper habitats associated with reefs. Many coral reef-associated fish species have highly specialized habitat requirements. Some species are typically found in sandy patches while some other will use different types of coral as shelters, which will lead to differing species assemblages

(*Bacchet, Zysman & Lefèvre, 2006*). Vocal species from these assemblages should create differential acoustic signatures in the frequency range in which they produce sounds. From this perspective, the objective of the present study was to investigate the variations of sound pressure levels in the low frequency range between underexplored habitats adjacent to coral reefs on the north coast of Moorea Island, French Polynesia (17°30′S, 149°5′W) and thereby provide a first insight into the acoustic features of these biotopes.

## MATERIAL & METHODS

### Study sites

The study was carried out at the end of the warm season, from June to July 2015, along three North-orientated seaward transects characterized by increasing depths, extending from the barrier reef (BR, <20 m), to the sandy plain (SP, 30–50 m), the reef slope (RS, 50–65 m) and to the more distant reef drop-off (DO, 75–100 m). The barrier reef of transect 1 was located in a Marine Protected Area (MPA) while barrier reefs of transects 2 and 3 were located in non-protected areas. Transects 1 and 2 were separated by the pass of Tiahura, a coral-free area located in front of an opening in the reef crest that canalizes the water flow to the ocean (Fig. 1). *Bertucci et al. (2015)* showed that this habitat had a mean sound intensity of ca. 90 dB re 1 μPa (20–5,000 Hz range). Transects 1 and 2 were 1.0 km apart, transects 2 and 3 were 1.3 km apart. For each transect, four different habitats corresponding to different depths were explored from the barrier towards the ocean: the barrier reef (BR, characterized by a water depth of 1–20 m and a substratum comprised of up to 40% live coral and a wide range of fish and invertebrate species; the sandy plain (SP), constituting the base of the reef with a declivity of 30–45° and characterised by vast expanses of patchy rocks covered by coral, with a high species diversity and located at 30–50 m depth; the reef slope (RS), characterized by a change of slope with an increased declivity located at 50–65 m depth, this zone of sedimentary accumulation is remarkable for its low specific richness and the density of benthic communities; and the reef drop-off (DO), characterized by a cliff located at 75–100 m depth and numerous fish species. Depth and topography were measured with a multi-beam sonar (Lowrance LMS 527) installed on an 18-feet boat. The positions of the different recording sites were localized with a GPS in order to replicate measurements.

### Recordings

Recordings were conducted between 09:00 and 16:00. This period of time shows little variations in acoustic activity, in contrast to early morning and late afternoon where drastic changes in sound intensity and complexity may happen rapidly (*Bertucci et al., 2015*; *Bertucci et al., 2016*). Recordings were made when wind speed was lower than 5 knots, with no swell, so as to prevent the boat from drifting and bobbing, and to reduce noise from the wind and sea surface turbulence against the boat's hull. Recordings were only made when no other boats were observed in the recording area.

An underwater Remora acoustic recorder (Loggerhead Instruments, Sarasota, FL, USA) connected to a HTI96-min hydrophone (sensitivity: −211 dB re: 1V for a sound pressure of 1 μPa, frequency response: 2 Hz–30 kHz; High Tech Inc., Long Beach, MS, USA) was

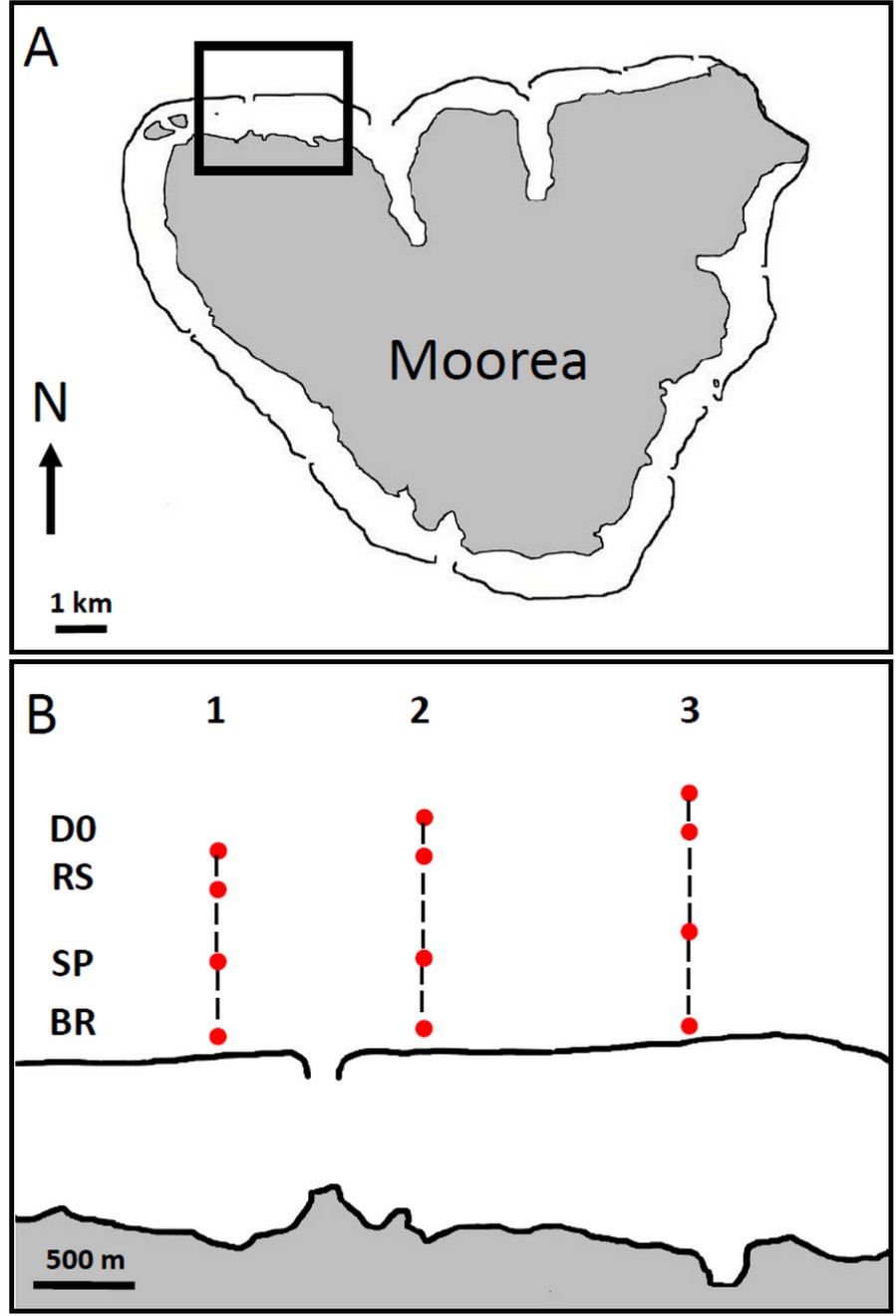

**Figure 1   Maps of Moorea Island (A), showing the locations of the recording sites along the 3 transects on the North Coast (B).** BR, barrier reef; SP, sandy plain; RS, reef slope and DO, reef drop-off. Maps drawn by the authors from an aerial photograph of Moorea taken by the CRIOBE in 2008 from a private plane. Surrounding black lines along the coasts of the island indicate barrier reefs.

**Table 1  Summary of the date, hour and depth of the recordings performed in each habitat type for the three different transects.**

| Transect | 1 | | | 2 | | | 3 | | |
|---|---|---|---|---|---|---|---|---|---|
| Habitat | Day | Hour | Depth (m) | Day | Hour | Depth (m) | Day | Hour | Depth (m) |
| BR | May 17 | 09:40 | 15 | May 20 | 10:55 | 20 | May 18 | 14:00 | 15 |
| | June 3 | 11:20 | 15 | June 6 | 09:40 | 15 | June 5 | 15:50 | 20 |
| | June 14 | 15:30 | 20 | June 25 | 14:05 | 15 | June 14 | 11:00 | 15 |
| SP | May 17 | 10:00 | 30 | May 20 | 11:15 | 30 | May 18 | 14:20 | 40 |
| | June 3 | 11:00 | 40 | June 6 | 09:20 | 40 | June 5 | 15:30 | 45 |
| | June 14 | 15:50 | 40 | June 25 | 13:45 | 35 | June 14 | 11:20 | 40 |
| RS | May 17 | 10:20 | 55 | May 20 | 11:35 | 65 | May 18 | 14:40 | 60 |
| | June 3 | 10:40 | 60 | June 6 | 08:30 | 65 | June 5 | 15:10 | 65 |
| | June 14 | 14:50 | 65 | June 25 | 13:25 | 60 | June 14 | 10:20 | 55 |
| DO | May 17 | 10:40 | 75 | May 20 | 11:55 | 75 | May 18 | 15:00 | 90 |
| | June 3 | 10:20 | 80 | June 6 | 08:50 | 80 | June 5 | 14:50 | 75 |
| | June 14 | 15:10 | 80 | June 25 | 13:05 | 70 | June 14 | 10:40 | 80 |

**Notes.**
BR, barrier reef; SP, sandy plain; RS, reef slope; DO, reef drop-off.

attached to a block of lead placed at the end of a 100 m rope. This minimized vibration of the rope and current-driven movement of the device. The measurements sequence consisted of recording the four different habitats of a single northward transect, i.e., BR, SP, RS, DO, in a random order. For each habitat, the recorder was suspended from the boat and lowered by an experimenter into the water until it was 5 m above the sea floor. The depth of the device (recorder and hydrophone) was determined by means of marks positioned every 5 m along the rope. Water depth was measured every 2 min with the sonar system to ensure that recordings were made within the appropriate habitat, at a constant depth, and that the hydrophone did not risk hitting the sea floor. Recordings lasted 10 min (sampling rate of 44.1 kHz, 16-bit resolution with a 33 dB gain) before the recorder was pulled back on board and switched off. Completing one transect took approximately 90 min. For each transect, three replicates were obtained for each habitat type with a one week time interval between them (Table 1). A total of 360 min of recordings were collected.

## Data analysis

A 20 Hz high-pass frequency filter was applied to all recordings to eliminate very low frequencies. The start and end sections corresponding respectively to the positioning and withdrawal of the recorder were deleted. Recordings were further cleansed by visually inspecting sound spectrograms using Avisoft SASLab Pro 5.2.07 software (Avisoft Bioacoustics, Glienicke, Germany) in order to cut-out anthropogenic sound sources and other artefactual sounds (e.g., animals probing the recording device or movement of the rope). This cleansing-step shortened some of the recordings down to 4 min. For each habitat, a set of 12 subsamples of 60 s were used, which were randomly extracted from the 3 replicates in order to produce spectra based on recordings of the same duration (Fast Fourier Transform FFT, 1,024 points Hamming window, providing a 21.53 Hz resolution). The sound pressure levels measured for each 21.53 Hz frequency band (SPL in dB re:

1 μPa, logarithmic scale) were transformed into μPa (linear scale) and averaged. Averaged sound pressure levels were then converted back into dB re: 1 μPa to present the average spectrum of each habitat. The characteristics of each habitat were described on the basis of variations of the average spectral profiles. For each habitat, the root mean square (RMS) of the sound pressure level was measured on the 20 Hz–2.5 kHz frequency band using Avisoft SASLab Pro 5.2.07 software. This low frequency band is dominated by fish vocalizations (*Lobel, Kaatz & Rice, 2010*; *Tavolga, Popper & Fay, 2012*). Due to a low sample size (three replicates for each sampling point), average sound pressure levels were compared between the three transects and also between the four different habitats with Kruskal–Wallis tests, followed by Tukey's post-hoc tests for pairwise comparisons. Intensity values of each frequency bins ($N = 116$) of power spectra were normally distributed (Shapiro–Wilks tests, $W = 0.98 - 0.99$, all $P > 0.05$) and were compared between the three transects and the four habitat types with two-way ANOVAs followed by Tukey's post-hoc tests for pairwise comparisons.

All analyses were two-tailed, at $\alpha = 0.05$ and carried out with R 3.1.2 (*R Core Team, 2014*) using customized scripts.

## RESULTS

### Spectral signatures of deeper habitats are mainly characterized by lower sound intensities

For each transect, BR locations presented spectra with significantly higher average sound pressure levels, followed by SP, RS and DO. Average sound pressure levels decreased from BR towards the most distant and deepest habitat, i.e., DO (Kruskal–Wallis, $\chi^2 = 19.24 - 30.62$, $df = 3$, all $P$-values $< 10^{-3}$; Tukey's post-hoc tests for pairwise comparisons) (Fig. 2). For transect 1, BR showed significantly higher average sound pressure levels than the 3 other habitats for all frequencies above 100 Hz. SP showed significantly higher average sound pressure levels than RS and DO from 150 Hz to 2,500 Hz (only a narrow 2,200 Hz–2,300 Hz range did not differ significantly between SP and RS). RS and DO spectra were not significantly different for any frequencies but a narrow frequency band from 1,250 Hz to 1,400 Hz (Two-way ANOVA, $F_{3,116} = 2.00 - 2.70$; $P < 0.05$) (Fig. 3A). For transect 2, BR and SP differed significantly only from 400 Hz to 1,500 Hz. As for transect 1, RS and DO showed significantly lower intensities than BR for most frequencies, and were significantly lower in their intensities than SP for frequencies below 2,000 Hz. In contrast to transect 1, RS showed significantly higher average sound pressure levels than DO for frequencies below 750 Hz, between 1,400 Hz and 1,500 Hz, and from 2,000 Hz to 2,300 Hz (Two-way ANOVA, $F_{3,116} = 2.50$; $P < 0.05$) (Fig. 3B). For transect 3, all spectra but RS and DO were significantly different for all frequencies from 400 Hz to 2,000 Hz with decreasing intensities from BR to DO. BR showed significantly higher intensities at lower frequencies too and significantly higher intensities than RS and DO above 2,000 Hz. RS and DO did not differ significantly in their intensities except for a very narrow 1,100 Hz –1,200 Hz band caused by the absence of a peak present in the other spectra (Two-way ANOVA, $F_{3,116} = 2.30$; $P < 0.05$) (Fig. 3C).

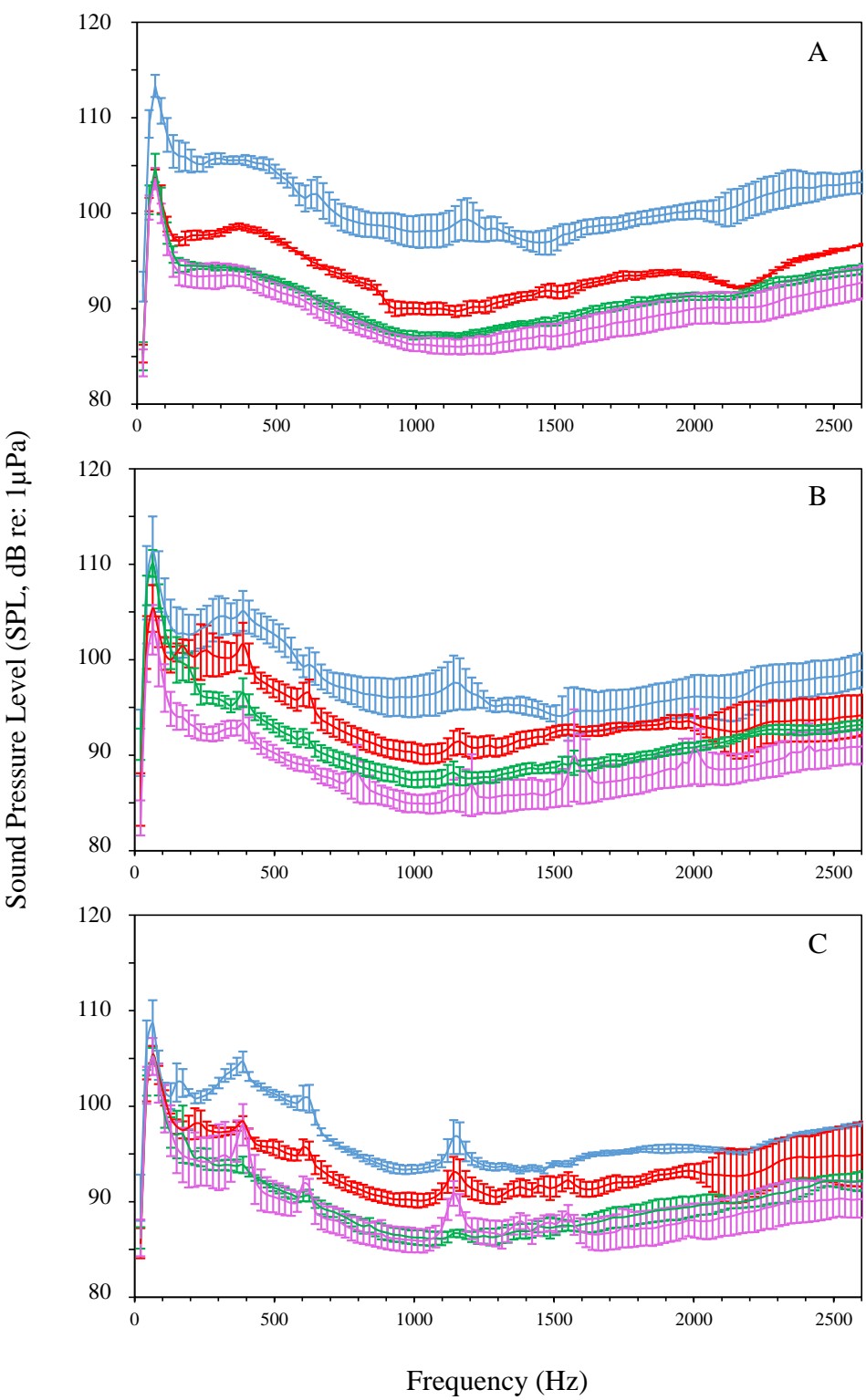

**Figure 2** **Average power spectra of the four different habitats recorded along transect 1 (A), 2 (B) and 3 (C) at the North Coast of Moorea.** Blue line, barrier reef (BR); red line, sandy plain (SP); green line, reef slope (RS) and purple line, reef drop-off (DO). The 20 Hz–2.5 kHz frequency band is presented. Values are mean ± S.D.

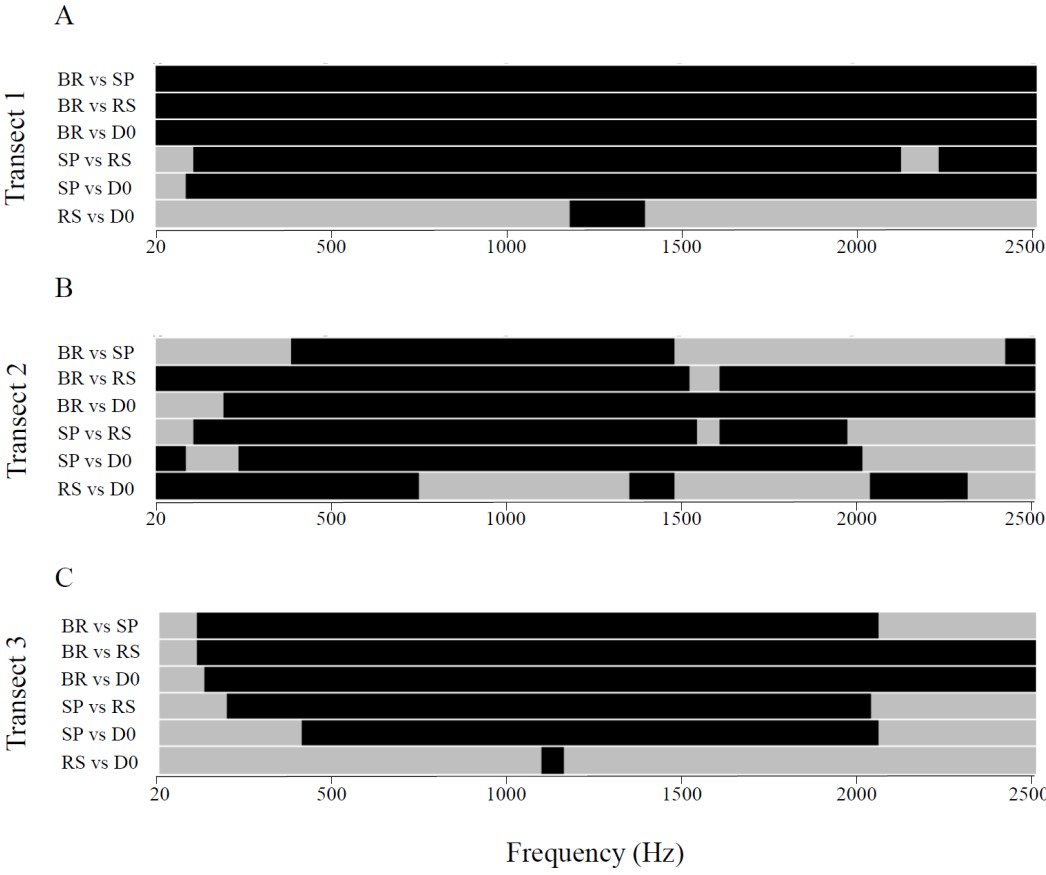

**Figure 3** **Between-habitat comparisons of sound intensities for the three transects in the 20 Hz–2.5 kHz frequency band.** Transects 1 (A), 2 (B) and 3 (C). BR, barrier reef; SP, sandy plain; RS, reef slope and DO, reef drop-off. Two-way ANOVA followed by Tukey's post-hoc tests for pairwise comparisons; Black: $P < 0.05$, Grey: non-significant difference.

## Similar habitats show differences in their spectral signatures

At the habitat type level, BRs and RSs showed the greatest difference in power spectra between the three transects. For BR, transect 1 showed significantly higher intensities for all frequencies of the spectrum compared to transect 2. The spectrum of BR of transect 2 was characterised by two intensity peaks around 600 Hz and 1,200 Hz. The spectra of transects 1 and 3 differed significantly for frequencies above 1,250 Hz. BR spectra of transects 2 and 3 differed significantly only for a narrow frequency range between 1,250 Hz and 1,500 Hz (Two-way ANOVA, $F_{2,116} = 2.80$ ; $P < 0.05$) (Fig. 4). For RS, transect 1 showed the significantly highest intensities for most frequencies above 500 Hz while transects 2 and 3 were similar (Two-way ANOVA, $F_{2,116} = 2.45$ ; $P < 0.05$) (Fig. 4). Spectra of SP and DO showed little variation in their intensities between the three transects. DO of transect 1 showed higher intensities below 400 Hz than DO of transect 2; and DO of transect 2 and 3 differed for higher frequencies between 1,600 Hz and 2,300 Hz (Two-way ANOVA, $F_{2,116} = 2.35 - 3.00$; $P < 0.05$) (Fig. 4).
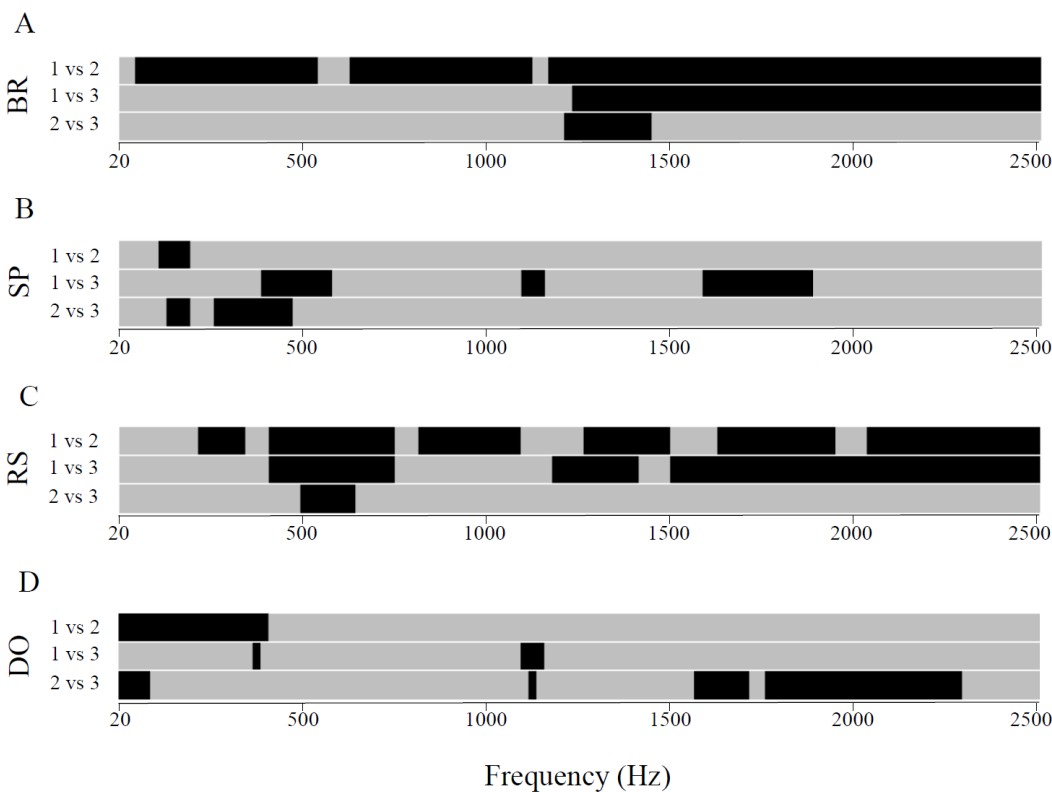

**Figure 4 Between-transect comparisons of sound intensities for each recorded habitat in the 20 Hz–2.5 kHz frequency band.** BR, barrier reef (A); SP, sandy plain (B); RS, reef slope (C) and DO, reef drop-off (D). Two-way ANOVA followed by Tukey's post-hoc tests for pairwise comparisons; Black: $P < 0.05$, Grey: non-significant difference.

The BR of transect 1 displayed the highest average sound pressure level (Kruskal–Wallis, $\chi^2 = 11.24$, $df = 2$, $P = 0.004$, Tukey's post-hoc tests for pairwise comparisons, all $P$-values $< 0.05$) (Fig. 5). No difference was observed between BRs of transect 2 and 3. The power spectra of SPs showed an inverted pattern with SP of transect 1 showing the lowest average sound pressure level (Kruskal–Wallis, $\chi^2 = 10.27$, $df = 2$, $P = 0.006$, Tukey's post-hoc tests for pairwise comparisons, all $P$-values $< 0.05$) (Fig. 5). No difference was observed between SPs of transect 2 and 3. RS and DO showed the significantly lowest sound pressure level for all transects (Kruskal-Wallis, $\chi^2 = 19.24 - 30.62$, $df = 3$, all $P$-values $< 10^{-3}$; Tukey's post-hoc tests for pairwise comparisons) with all values below 92 dB re: 1 μPa. RS of transect 2 showed significantly lower average sound pressure level (Kruskal-Wallis, $\chi^2 = 9.08$, $df = 2$, $P = 0.010$, Tukey's post-hoc tests for pairwise comparisons, all $P$-values $< 0.05$). The DO of transect 2 showed a significantly higher average sound pressure level of 95 dB re: 1μPa (Kruskal-Wallis, $\chi^2 = 13.14$, $df = 2$, $P = 0.001$, Tukey's post-hoc tests for pairwise comparisons, all $P$-values $< 0.05$) (Fig. 5).

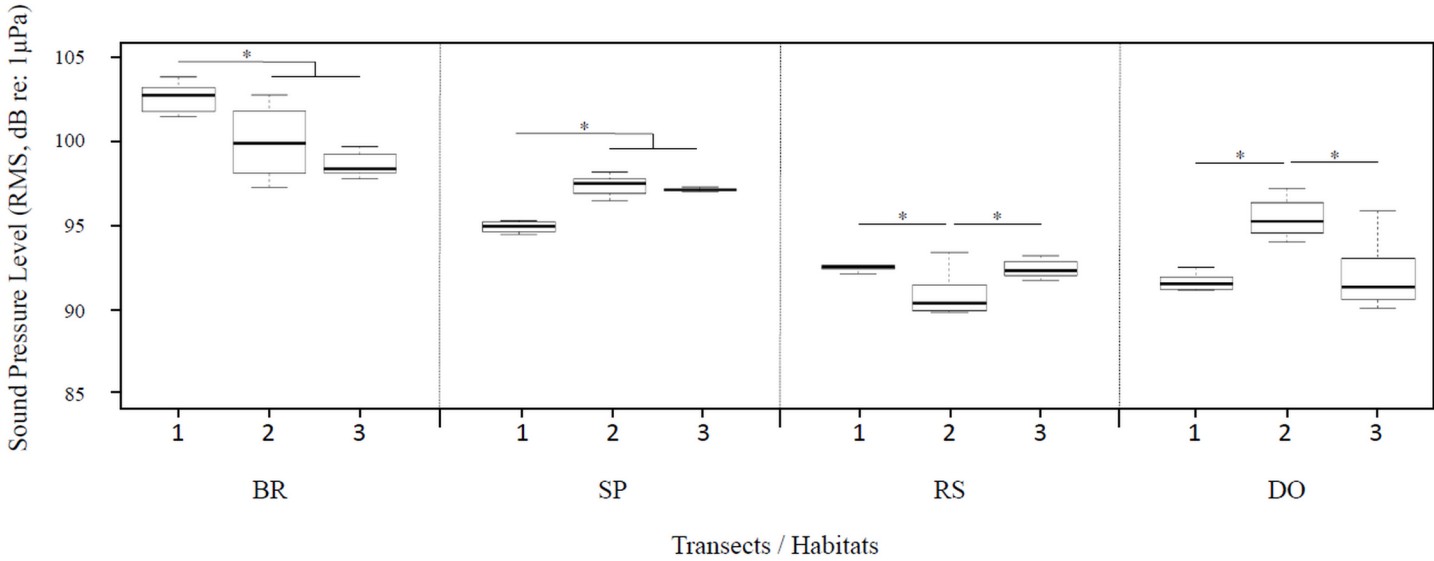

**Figure 5** **Ambient sound pressure levels (RMS values in dB re: 1 μPa) in the 20 Hz–2.5 kHz frequency band for each habitat along the three transects.** The four different habitats are separated by dashed lines. Boxes represent the first and third quartiles, thick horizontal bars are the median (second quartile) and whiskers correspond to the range (min–max) of the distributions. Kruskal–Wallis tests followed by Tukey's post hoc tests for pairwise comparisons. Asterisks show significant differences between transects at $P < 0.05$.

## DISCUSSION

Recent studies on coral reefs have highlighted the positive relationships between sound signatures and coral cover, density of fishes or increased number of biotic sound sources in shallow waters (*Kaplan et al., 2015*; *Nedelec et al., 2015*; *Bertucci et al., 2016*). In this study, comparison of the spectra of four types of habitats adjacent to coral reefs, characterized by different topographies and substrates, by increasing depths and distances from the reef, revealed different spectral profiles most probably related to their different physical and biological properties.

Recordings made at the barrier reef presented higher sound pressure levels in the low frequency range despite the fact that low frequencies transmit poorly in shallow waters (*Rogers & Cox, 1988*). The higher level of the biophony would suggest this habitat type is occupied by more vocal organisms than the three others. We cannot exclude, however, the possibilty that low frequencies are related to a greater contribution of the geophony at the barrier reefs, with sounds produced by crashing waves or by moving substrate (sand) leading to increased sound pressure levels. The short distance between the sea surface and the bottom can also produce more reverberation in shallow waters than in deeper waters and increase sound levels at higher frequencies. Hence, lower sound levels in deeper habitats do not necessarily mean that mesophotic reefs would be less acoustically rich than barrier reefs. A description of potential vocal species present in these habitats together with the description of their physical environments may help to distinguish between the respective contributions from different sources, i.e., biophony and geophony.

**Table 2 Recorded and predicted sound pressure levels (average values in dB re: 1 μPa) in the 20Hz–2.5 kHz frequency band for each habitat along the three transects.** Transmission loss (TL) was calculated using cylindrical spreading model as TL = 10 log R with R = distance from the source.

| Transect | Habitat | Horizontal distance from BR (m) | Depth (m) | Distance from source (m) | SPL (dB re 1 μPa) | | | |
|---|---|---|---|---|---|---|---|---|
| | | | | | Recorded | TL | Predicted | Difference |
| 1 | BR | 0 | 17 | 0 | 102.7 | – | – | – |
| | SP | 149 | 37 | 153 | 95.0 | 21.9 | 80.8 | 14.2 |
| | RS | 268 | 60 | 275 | 92.6 | 24.4 | 78.3 | 14.3 |
| | DO | 275 | 78 | 286 | 91.5 | 24.6 | 78.1 | 13.4 |
| 2 | BR | 0 | 17 | 0 | 99.9 | – | – | – |
| | SP | 154 | 35 | 158 | 97.5 | 22.0 | 77.9 | 19.6 |
| | RS | 340 | 63 | 346 | 90.4 | 25.4 | 74.5 | 15.9 |
| | DO | 345 | 75 | 353 | 95.2 | 25.5 | 74.4 | 20.8 |
| 3 | BR | 0 | 17 | 0 | 98.3 | – | – | – |
| | SP | 204 | 42 | 208 | 97.1 | 23.2 | 75.1 | 22.0 |
| | RS | 392 | 60 | 397 | 92.3 | 26.0 | 72.3 | 20.0 |
| | DO | 400 | 82 | 408 | 91.3 | 26.1 | 72.2 | 19.1 |

**Notes.**

BR, barrier reef; SP, sandy plain; RS, reef slope; DO, reef drop-off.

Recorded sound pressure levels in deeper habitats highlight that sounds do not result from the sole propagation and degradation of sounds produced at the level of the noisy barrier reef, as predicted by propagation models (*Mann et al., 2007*) (Table 2). Moreover, sound levels of sandy plains of transects 2 and 3 showed higher average sound levels than their counterpart of transect 1 despite being more distant from their respective barrier reefs. This clearly supports that barrier reef is not the only sound source and that deeper habitats possess their own sound sources. The weak variations between the sandy plain spectra may result from their topography and physical characteristics with large patchy rocky habitats and similar communities between sites. At reef slopes, more distinct signatures appear again with reef slope of transect 3 being as noisy as reef slope of transect 1 despite being located further away from the barrier. Finally, while the spectra of drop-offs of transects 1 and 3 show no significant variations with their respective reef slopes, drop-off at transect 2 was characterized by a higher sound pressure level compared to its reef slope, especially at the low frequencies. So, the decrease of sound intensities along transects appears to be variable and the spectral profiles of the different habitats show significantly increased intensities despite their distance to the barrier reef. These observations would reinforce the idea that the barrier reef is not the single sound source responsible of observed spectra and that additional sources such as soniferous species—that are not present at the level of the barrier or may differ in abundance at deeper habitats—may actively play a role in the sonic signature of deeper environments. In particular, as the transitions between the reef slopes and the reef drop-offs are very short for all transects, the observed differences (especially for transect 2) between spectra and the potential link with differential biological activity deserve to be investigated. Indeed, while the reef slope is a zone of sedimentary

accumulation with a low species richness, the reef drop-off houses numerous fish species. This might explain the increased intensities between these 2 types of habitats.

Overall, as in shallow waters, differences in spectral patterns seem to exist in deeper habitats and may therefore reflect different characteristics of the habitats, i.e., physical and/or biological. Acoustic cues produced by deeper habitats may therefore also be used in the orientation of marine larvae and may come into play earlier in the recruitment of coral reefs organisms. However, the short period of time sampled in this study remains insufficient to reliably characterize the different habitats and only provides initial information on the different acoustic signatures of deeper habitats. Fish vocal activity can change drastically at dusk or dawn and be more sustained at night with specific species vocalizing in specific time windows (*Pieretti et al., 2017*). Monitoring for longer time periods might highlight further differences between habitats at the diel scale. Moreover, sound production in fishes is often linked to social activities, such as courtship interactions and spawning events that will vary on a longer, seasonal scale. Long term recordings of deeper environments would then be necessary to capture the complete picture and identify acoustic differences (*Pieretti et al., 2017*).

## CONCLUSION

We still know very little about the acoustic ecology of deeper habitats adjacent to coral reefs. It has been assumed that deeper environments were less likely to be impacted by anthropogenic activities or by global change, and therefore may provide refuge areas for shallow reef species (*Bongaerts et al., 2010*). This hypothesis has gained a growing interest in the scientific community, but has been only tested at few locations for few species. In the future, the opportunity to have proxies of the ecological state of these refuge areas by means of soundscape analysis and linking their acoustic characteristics to their refuge potential may help to better judge the impact of global change and the influence of these adjacent ecosystems on coral reefs. Several studies have demonstrated that barrier reef sound attracts fish and crustacean larvae during settlement onto the reef (*Barth et al., 2015*). The present study represents a first step towards the acoustic investigation of deeper environments and suggests that deeper habitats could also play a role in the orientation of larval marine organisms.

Research in practically unexplored depths will undoubtedly bring new knowledge and tools that will be extremely valuable for the creation of Marine Protected Areas (MPAs), watershed management plans and the development of conservation plans for coral reefs as a whole, from shallow to deep water habitats.

## ACKNOWLEDGEMENTS

Authors would like to thank Franck Lerouvreur for his technical support. We also thank two anonymous reviewers and Dr. Nadia Pieretti for their comments on a previous version of this manuscript.

### Funding

This study was supported by a research grant of the Total Foundation awarded to David Lecchini. The funders had no role in study design, data collection and analysis, decision to publish, or preparation of the manuscript.

### Grant Disclosures

The following grant information was disclosed by the authors:
Total Foundation award.

### Competing Interests

The authors declare there are no competing interests.

### Author Contributions

- Frédéric Bertucci conceived and designed the experiments, performed the experiments, analyzed the data, wrote the paper, prepared figures and/or tables, reviewed drafts of the paper.
- Eric Parmentier and David Lecchini conceived and designed the experiments, wrote the paper, reviewed drafts of the paper.
- Cécile Berthe and Marc Besson performed the experiments, reviewed drafts of the paper.
- Anthony D. Hawkins and Thierry Aubin wrote the paper, reviewed drafts of the paper.

### Data Availability

Bertucci, Frédéric (2017): Transect 1 Tiahura. figshare.
https://doi.org/10.6084/m9.figshare.5494399.v1
Bertucci, Frédéric (2017): Transect 2 Passe. figshare.
https://doi.org/10.6084/m9.figshare.5498041.v1
Bertucci, Frédéric (2017): Transect 3, Papetoai. figshare.
https://doi.org/10.6084/m9.figshare.5498047.v1.

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
