# Peer review of "Snapshot recordings provide a first description of the acoustic signatures of deeper habitats adjacent to coral reefs of Moorea"

_PeerJ, doi:10.7717/peerj.4019_

## Round 0.1 · original submission · Major Revisions

All three reviewers provided useful and detailed comments and suggestions. In particular, please consider Reviewer 2's concerns about the study design and interpretation. And whether the raw data files will be made available.

Reviewer 1 ·

Basic reporting

The article is clearly written and provides appropriate references. The figures make sense, although I would like to have supplemental material with a few acoustic recordings so that we could listen to the different soundscapes.

Experimental design

I find 2 major flaws with the design which limit the conclusions one can draw from the study:

1) Temporal window: I am concerned that recordings were only made during daytime hours. I think this is a big oversight. You mention that there is high variability during nighttime and early morning hours – but this is exactly when you may see larger differences between habitats. For example, if a species of soniferous fish lives only in deeper water, but they only sign at night, then you would not pick up this difference between depths/habitats by missing their prime singing window.

2) Temporal replication: I understand that perhaps logistical or financial constraints were at play here, and the investigators may not have had access to longer-term recorders or the ability to dive in deeper waters. But 10-minute recordings during subsequent mornings is not very high replication. I think the authors should emphasize the “snapshot” nature of these recordings and put this caveat in their results and possibly in the title.

That said, there is still value in this work, but it doesn't knock your socks off in terms of novelty or a grand contribution to the field.


There are also a few points of clarification needed:
-exact water depth of recordings (in a table)
-why no inclusion of higher frequency sounds
-how the spectral averaging was done
-did you look/listen to recordings and get any explanation for some of these differences/sound sources?


One other point about experimental design/analysis:
Did you try using any of the newer eco-acoustic indices like acoustic complexity, acoustic dissimilarity, etc? I am guessing that you may have tried, but found no significant patterns or perhaps even misleading results. You would not be alone – several others have tried and have learned that these indices are not “ready” for the marine environment yet. There was a lively discussion about this at a recent ASA meeting, and a recent paper by Staaterman et al (2017, MEPS) demonstrates an example in which the indices don’t work as expected. I know many authors are reluctant to put null results or confusing results into a manuscript, but I would urge you to include these analyses anyway. It would bring further emphasis to the fact that these indices require further ground-truthing for the marine environment.

Validity of the findings

Some of the interpretation of the differences between habitat types and distances from shore need to be refined. There needs to be a discussion about the propagation of low-frequency sounds in shallow water.

Otherwise, the conclusions are well-stated and reasonable.

Additional comments

Given the fact that the authors probably cannot go back and redo these recordings, or perhaps were limited with their instrumentation, it's not reasonable to ask them to increase replication. This is a small study but still a valid contribution to the field. For this reason I am putting "minor revisions" rather than "major revisions". But I would like to see them try running the newer eco-acoustic indices (with existing data), and to clarify about some of their methods for spectral averaging etc. And I encourage them to think about higher spatio-temporal resolution in future work, if time and money allows.


small comments here:

Introduction – line 48: you start two sentences in a row with “as an example”

Methods: - line 93- is Tiahura a natural canal or man-made? Why are there no corals there?

Line 110: I am concerned that recordings were only made during daytime hours. I think this is a big oversight. You mention that there is high variability during nighttime and early morning hours – but this is exactly when you may see larger differences between habitats. For example, if a species of soniferous fish lives only in deeper water, but they only sign at night, then you would not pick up this difference between depths/habitats by missing their prime singing window.

Please give more specifics about the water depths where recordings were made. I see that you have the range of water depth for each habitat type, but this range is pretty big (~20m).

Methods: so the boat was drifting, not anchored? To me this is another potential source of error. I have made recordings of the hull of a boat and the slapping/lapping of waves is indeed a problem. Were you not able to deploy passive recorders for longer periods?? How did you deal with these lapping noises?

Why did you not include analyzes of higher frequencies (snapping shrimp band)? I understand perhaps more interest in looking at fish vocalizations, but it would not be hard to also include snapping shrimp here.

I understand that perhaps logistical or financial constraints were at play here, and the investigators may not have had access to longer-term recorders or the ability to dive in deeper waters. But 10-minute recordings during subsequent mornings is not very high replication. I think the authors should deemphasize the “snapshot” nature of these recordings and put this caveat in their results and possibly in the title.

Lines 143-152: please clarify language here. It is not clear how you got the 12 samples. Also, please explain the “averaged for each sampled frequency” – did you not have power spectral density in 1-Hz bands? From the way your axes on the plots are labeled, it appears that you did not have 1-Hz bins and some averaging across frequencies took place. Please explain what bandwidths were used. Does the recorder automatically spit out SPLs and not the raw waveforms? This part needs clarification.

Line 163: I find the phrase “distant habitat” misleading here. You mean distance from shore, I assume, but you may also mean distance from other habitats. I would instead say “deeper habitats”. If you don’t mean distance from shore, this would imply that your hydrophones were not truly “in” the environment you claim to be sampling.

Lines 222-234: I think some discussion is required about the propagation of low-frequency sounds in very shallow water. Since transmission of these sounds is restricted by water depth, you would actually expect to have lower sound levels in these lower frequencies for the shallowest sites. The fact that you got the opposite demonstrates that there is a greater contribution from geophony or biophony.


Line 235-237: I don’t see how your results lead you to this conclusion. Higher sound levels at different habitats along transect 1 could mean more sound-sources at the other (non-reef) habitats, which you state later in the paragraph. I would remove this sentence or rephrase it. I would also mention later in this paragraph the fact that deep water enables transmission of lower-frequency sounds, so in addition to the fact that there may be additional sound sources in the reef slope/sandy area (fish), it may also have to do with the physics of the environment.

Did you try using any of the newer eco-acoustic indices like acoustic complexity, acoustic dissimilarity, etc? I am guessing that you may have tried, but found no significant patterns or perhaps even misleading results. You would not be alone – several others have tried and have learned that these indices are not “ready” for the marine environment yet. There was a lively discussion about this at a recent ASA meeting, and a recent paper by Staaterman et al (2017, MEPS) demonstrates an example in which the indices don’t work as expected. I know many authors are reluctant to put null results or confusing results into a manuscript, but I would urge you to include these analyses anyway. It would bring further emphasis to the fact that these indices require further ground-truthing for the marine environment.


A note for future work: I have been to Moorea and noticed the big cruise ships that come through. I would be curious to know how the propagation of the ship noise differs depending on geographic location, again due to the physical constraints of shallow vs. deep water. I would suggest you put out some passive recorders at different depths/transects along the reef slope and see what happens when a ship comes through. Would also be interesting to put some inside those two bays/cracks in the island.


Figure 5: could be displayed better – consider using color and putting all on one plot, or adding another panel with average for each habitat type.

Would be nice to have supplemental data with recordings from each habitat type.

Reviewer 2 ·

Basic reporting

Intro and background are sufficient to outline the problem. In general, the manuscript would benefit from review from a native English speaker for improved clarity and word use. A summary table of the raw data was supplied (although the raw data itself, i.e., wave files) was not. I am not sure if this conforms to PeerJ’s standards. Figures 3-4 are non-normative and I am not sure what they add in terms of interpretation.

Experimental design

I have three major problems with this study. Data collection during the day is obviously totally insufficient because this is not, as the authors note, the time when most soniferous organisms on reefs are active. Furthermore, it seems from the methods as if the transects were conducted sequentially and within transects the stations were visited south to north. Then this was replicated in the same way two more times. If that is true then there is a major bias there in terms of the times at which each station and transect were sampled. Finally, the authors attempt to compare the recordings they made to habitat type but bring no evidence of the fauna resident in each habitat type. For these three reasons, I cannot recommend publication of this research.

Validity of the findings

For the reasons stated above, I do not interpret these findings as valid.

Additional comments

Lines 82-83: “highly specialized habitat requirements” – such as?
Lines 82-85: the logical link between “highly specialized habitat requirements” and variations in low-frequency SPL is not clear. Suggest rephrasing.
Line 94: “noisy” – first of all this is somewhat subjective, but 90 dB is not, in my opinion, “noisy”, or, more appropriately, high amplitude
Lines 98-105: could the authors please bring evidence to support their claims about the species richness of each of these habitat types. Did they conduct visual surveys? Or is this conjecture?
Lines 131-133: The close to shore station was always visited first in each transect and replicate? If sound levels change throughout the day then this would introduce a major bias into these results. Did the authors not put recorders at each station for the entire day or ideally over several diel periods? If not, there is no way to know how sound levels were changing during the sampling period.
Results section: I found this entire section very hard to understand.
Lines 231-234: True, it is very important to consider the physical environment and its influence on ambient sound. Because the authors do not have visual survey data to link to fauna at each reef, it seems as if purely physical differences among the stations would be the most parsimonious explanation of the differences found.
Lines 258-262: I agree with the statement here that the short sampling period is insufficient to properly characterize these habitats. A large body of literature now exists that points to the importance of diel sampling because of how dramatically sound levels can change even over short temporal scales.

·

Basic reporting

no comment

Experimental design

no comment

Validity of the findings

no comment

Additional comments

The Ms provides a first insight into the acoustic features of different habitats adjacent to the coral reef surrounding the north coast of Moorea (French Polynesia). In particular, it focuses on the description of soundscape signatures (sound energy along frequencies) along three transects of four recorded points each (barrier reef; sandy plain; reef slope; drop-off).
The Ms is well organized and the results produce interesting advance to our knowledge of the field. These baseline data are strongly needed to produce a benchmark that could be used as a reference for future biomonitoring studies. Here I provide a list of recommendations to improve the MS.

The authors have provided a good temporal and spatial diversification of samples by investigating three points for every habitat, and replicating in three different days. Anyway, the total amount of recorded minutes in each of them is quite limited. Soundscape holds a great deal of variation over time of the day and over days. It could substantially change either to the time of the day in which recordings were taken (for instance many fishes vocalize in specific temporal windows) or may also vary from one day to the next (see Pieretti et al. 2016). In order to capture a reliable example of the soundscape of the area, longer recordings would be the optimum solution.
The authors’ dataset is limited to 12minutes (divided in 4mins in three replicates) for every recording station. That makes 36 minutes for every habitat type. Moreover, authors do not fully specify the recording time of the day (line 110: Recordings were conducted between 09:00 and 16:00).
I still consider the collected data a useful and needed starting reference for future studies, however in the discussion I suggest to explain this limitation (adding details at line 261) or to the reasons that have led authors to consider it as less important (maybe in these tropical environments soundscape fluctuations are weak).

I recommend that a native English-speaking colleague review the Ms. In fact, examples of some errors and typos along the paper are at line 93 (depths), 110 (recordings), 181 (did not differed) or the sentence at line 101-103 should be rephrased.

The statistics should be better presented and clearly explained. At lines 153-160, the current paragraph makes its understanding challenging. I suggest specifying the number of cases (N) - embedded in the text or in the caption of figures – so that the reader can easily understand which data have been considered for each analysis.

Others:
Keywords are missing.
Line 143: FFT 1024 on the 48kHz? (line 130) Figures seem to have a higher resolution in frequencies.
Line 143-144: I would explain why this step was taken
Line 173: I would refer to figures before
Line 179: from
Lines 224-226: it was not stated earlier
Figure 1: rescale the distance among transects; the distance among 2 and 3 seems to double the distance of transect 1 and 2
Figure 5: Caption: refer to the statistics used


Dr. Nadia Pieretti
Dipartimento Scienze Pure e Applicate - DiSPeA
Università degli Studi di Urbino "Carlo Bo"
Campus Scientifico Enrico Mattei
Località Crocicchia
61029 Urbino
Italy

---

## Round 0.2 · accepted · Accept

Thank you for carefully responding to all of the suggested edits and comments from the reviewers and for editing your paper accordingly. I have carefully read your manuscript and made several minor suggested edits (see attached MS Word file converted to a PDF) which you could consider while in Production.